# Meconium and Amniotic Fluid IgG Fc Binding Protein (FcGBP) Concentrations in Neonates Delivered by Cesarean Section and by Vaginal Birth in the Third Trimester of Pregnancy

**DOI:** 10.3390/ijms26157579

**Published:** 2025-08-05

**Authors:** Barbara Lisowska-Myjak, Kamil Szczepanik, Ewa Skarżyńska, Artur Jakimiuk

**Affiliations:** 1Department of Biochemistry and Pharmacogenomics, Medical University of Warsaw, 02-097 Warsaw, Poland; s080965@student.wum.edu.pl; 2Department of Laboratory Medicine, Medical University of Warsaw, 02-097 Warsaw, Poland; ewa.skarzynska@wum.edu.pl; 3Department of Obstetrics, Women’s Diseases and Gynecologic Oncology, National Medical Institute of the Ministry of the Interior and Administration, 02-507 Warsaw, Poland; jakimiuk@yahoo.com; 4Center for Reproductive Health, Institute of Mother and Child, 01-211 Warsaw, Poland

**Keywords:** meconium, fetus, FcGBP, amniotic fluid, third trimester of pregnancy, cesarean section

## Abstract

IgG Fc binding protein (FcGBP) is a mucin-like protein that binds strongly to IgG and IgG–antigen complexes in intestinal mucus. FcGBP presence and its altered expression levels in meconium accumulating in the fetal intestine and amniotic fluid flowing in the intestine may provide new knowledge of the mechanisms responsible for the immune adaptation of the fetus to extrauterine life. FcGBP concentrations were measured by ELISA in the first-pass meconium and amniotic fluid samples collected from 120 healthy neonates delivered by either vaginal birth (*n* = 35) or cesarean section (*n* = 85) at 36 to 41 weeks gestation. The meconium FcGBP concentrations (405.78 ± 145.22 ng/g) decreased (r = −0.241, *p* = 0.007) over the course of 36 to 41 weeks gestation, but there were no significant changes (*p* > 0.05) in the amniotic fluid FcGBP (135.70 ± 35.83 ng/mL) in the same period. Both meconium and amniotic fluid FcGBP concentrations were higher (*p* < 0.05) in neonates delivered by cesarean section. Decreases in the meconium FcGBP concentrations correlated (r = −0.37, *p* = 0.027) with the gestational age in neonates delivered by vaginal birth but not in those delivered by cesarean section (*p* > 0.05). No association was found between the FcGBP concentrations in meconium and amniotic fluid and the birth weight (*p* > 0.05). With the development of the mucosal immune system in the fetal intestine over the course of the third trimester of gestation, the meconium FcGBP concentrations decrease. Increased FcGBP concentrations measured in the meconium and amniotic fluid of neonates delivered by cesarean section may possibly indicate altered intestinal mucosal function. Intrauterine growth is not associated with the intestinal mucosal barrier maturation involving FcGBP.

## 1. Introduction

IgG Fc binding protein (FcGBP) is a mucin-like protein produced by and released from goblet cells of the human intestine and characterized by its individual ability to bind to IgG and IgG–antigen complexes. Mucins, a family of high molecular weight glycoproteins (>200 kDa), are structural components of mucus [1,2,3].

During its development in utero, a fetus is exposed to a wide variety of commensal and pathogenic bacteria as well as food antigens [4]. The intraluminal mucosal immune system of the intestine must be appropriately balanced to provide protection against pathogens potentially harmful to the gastro-intestinal tract while enabling education of the immune system by the specialized transport of antigens across the intestinal epithelium, as these specific immune responses are critical for fetal development [4,5,6]. It is still unclear whether the characteristic ability of FcGBP to form complexes with IgG could be involved in inhibiting the transport of IgG and its complexes with antigens from the intestinal lumen to the antigen-presenting cells.

Abundant transplacental transport of maternal IgG provides the fetus with high levels of IgG and its complexes formed with specific antigens [7,8]. Maintaining homeostasis in the fetal intestinal lymphatic system enables correct responses and tolerance of antigens supplied to the fetal intestine in utero and adaptation of the fetus to life after birth [9]. Limiting the interaction of antigens with the intestinal immune system by binding of IgG and IgG–antigen complexes to FcGBP may protect the fetus from harmful compounds but also inhibit the innate mechanism of presentation and initiation of adaptive immune responses in the regional lymphatic structures [4,10,11].

According to Kristensen [12], neonates delivered by cesarean section are a group at risk of disease associated with mucosal immune function. It is assumed that mucosal immune dysfunction in the fetal period in neonates delivered by cesarean section may be responsible for the origin and development of immune diseases after birth [7,12,13].

Meconium and amniotic fluid are easily accessible biological materials that are in direct contact with the fetal intestine lumen and potentially may provide new knowledge about the development of the fetus [14,15].

The aim of the study was to measure FcGBP concentrations in the amniotic fluid and corresponding meconium samples obtained from neonates delivered by cesarean section and by vaginal birth at 36 to 41 weeks gestation.

## 2. Results

The FcGBP concentrations in the corresponding samples of meconium and amniotic fluid collected from 120 neonates born at 36 to 41 weeks gestation are presented in Table 1.

A small positive correlation was found between the FcGBP concentrations in the meconium and in the amniotic fluid (r = 0.18, *p* = 0.053).

The effects of gestation length within the last 36 to 41 weeks on variations in FcGBP concentrations differ between the meconium and the amniotic fluid. The coefficient of correlation between week gestation and FcGBP concentration in the meconium was negative and statistically significant (r = −0.25, *p* = 0.007), while in the amniotic fluid it was also negative but not statistically significant (r = −0.14, *p* = 0.129).

Figure 1 shows variations in the meconium FcGBP concentrations found for individual neonates over the course of 36 to 41 weeks gestation.

The scatter graph (Figure 1) confirms falls in the meconium FcGBP concentrations over the course of the third trimester with several observations of a dispersion in the concentrations measured week-by-week.

Table 2 compares the effect of the delivery method on the FcGBP concentrations in the meconium and amniotic fluid.

Neonates delivered by cesarean section demonstrated significantly higher (*p* < 0.05) FcGBP concentrations in both meconium and amniotic fluid.

The graph in Figure 2 compares the variations in the meconium FcGBP concentrations over 36 to 41 weeks gestation found in neonates delivered by vaginal birth and by cesarean section.

A significant negative correlation (r = −0.37, *p* = 0.027) was found between the meconium FcGBP and pregnancy length week-by-week for the neonates delivered by vaginal birth, but no significant correlation was established for those born by cesarean section (r = −0.08, *p* = 0.459).

No effect was found of the gestation length within the period of 36 to 41 weeks on the variations in the amniotic fluid FcGBP concentrations for both neonates delivered by vaginal birth and those born by cesarean section (the correlation coefficients for FcGBP vs. weeks gestation: r = −0.22, *p* = 0.211, r = 0.00, *p* = 0.995, respectively).

No association was established (*p* > 0.05) between the meconium and amniotic fluid FcGBP concentrations and the birth weight (Table 3).

There were no significant differences (Mann–Whitney U test) in FcGBP concentrations in meconium and amniotic fluid between boys (*n* = 62) and girls (*n* = 58), *p* > 0.05.

## 3. Discussion

The findings of this study are the first to demonstrate the presence of FcGBP in meconium and amniotic fluid and suggest the potential uses of FcGBP determinations to assess the development of the intestinal mucosal immunity in the fetus.

FcGBP widely expressed by epithelial tissues is one of 13 distinguished types of mucins produced by goblet cells of the human intestine [1]. A complex network made up of components of intestinal mucus forms a barrier that protects the intestinal epithelium from exposure to bacteria and other foreign antigens entering the intestinal lumen. FcGBP, via an additional mechanism of binding strongly to IgG and IgG–antigen complexes with antigens in intestinal mucus, could prevent the interaction of antigens with the surface of intestinal epithelial cells. The function of FcGBP in the immune system differs from that of Fc gamma receptors, and an independent role for FcGBP is suggested in the protection of the fetal intestinal mucosa [1,2,3]. IgG is the only antibody class able to be transferred across intestinal epithelial cells during the maturation of the fetal intestinal immune system [7,8].

Decreasing meconium FcGBP concentrations measured week-by-week in the third trimester may confirm the Developmental Origins of Health and Disease (DOHaD) hypothesis, which suggests developmental plasticity of the fetus dependent on the individual external and internal environment [4,9]. The fetal circulation IgG levels continue to increase from 17–41 weeks of gestation with a sharp increase after week 36 and usually exceed maternal IgG concentrations by 30% at full term [8,16]. Swallowed amniotic fluid also contains IgG. IgG and antigens bound to IgG potentially may be transferred to the fetal circulation across the intestinal epithelial cells to the specialized immune cells via the neonatal Fc receptor (FcRn), which has a key function in the intracellular transport [17]. The decreasing meconium FcGBP concentrations in the third trimester reported in this study are thus opposed to the increasing levels of transferred maternal IgG. Binding of IgG and IgG–antigen complexes by FcGBP reduces the possibility of their interaction with the FcRn responsible for antigen transport to the immune cells [16,18].

A question arises whether consistently decreasing meconium FcGBP concentrations observed in the third trimester may reflect the increasing fetal demand for IgG or whether they are the effect of complex regulatory mechanisms that control the interaction between the fetal intestinal mucosa and antigens. It seems rational to assume that reduced FcGBP levels facilitate the uptake of antigens from the intestinal lumen by FcRn, which may be a biological effect developed to initiate the adaptive immune response in the regional lymphatic structures of the fetal intestine [19,20].

Making antigens available for the activation of immune cells in the fetal intestine is the essential task of the immune system education to recognize and differentiate between self and non-self antigens, safe nutrient proteins, commensals, and pathogens [4,7,21,22]. The actual antigen type and its biological significance for the response of luminal immune cells remain unclear. The intrauterine environment of the developing fetus was long considered sterile, but now an increasing number of studies confirm direct effects of maternal microbiota on the fetal and neonatal development [4,23]. There have been numerous attempts documented in the literature [4,6,10,14,24] to elucidate the impact of maternal microbiota and food allergens on later stages in the fetal development and health in the postnatal period. Researchers have not demonstrated the presence of live bacteria in the placenta, but it has been suggested that maturation of the fetal intestinal mucosal immunity may rely on the transport of microbial metabolites and microbial fragments across the fetal intestinal epithelial cells with the participation of maternal IgG [25].

It is generally agreed that maternal-driven education of the fetal immune system is beneficial, as it enhances immune tolerance in offspring and may be the main factor modifying immunity and lowering the risk of later development of allergies. One study was performed to assess whether there is a preferential period during pregnancy that provides the optimal immune response to maternal antigens in the fetus and what effects it has for postnatal development [26]. Our observations of the decreasing meconium FcGBP concentrations over the course of the third trimester of pregnancy suggest that antigen presentation to immune cells is gradually made possible in line with the maturation of the fetal intestine. Interestingly, the meconium FcGBP values were more dispersed, which suggests possible individual effects produced by different antigens in the intestinal mucosal immune system at different stages of its development.

The findings of the study did not show any association between any immune alterations in the fetal intestine involving FcGBP and increased birth weight.

As cesarean section rates continue to increase, there are justified concerns about proper adaptation of neonates born by cesarean section to perinatal exposures of postnatal feeding and pathogens. Some authors [7] suggest the mucosal immune dysfunction in the fetal period may increase the postnatal risk of autoimmune diseases. This hypothesis is confirmed by observations in a large sample of children aged 0–14 years born by cesarean section, which demonstrated an increased risk of disease associated with immune function like asthma, laryngitis, and gastroenteritis [12,13]. The present study found significantly higher FcGBP concentrations in both meconium and amniotic fluid from the neonates born by cesarean section. Additionally, unlike the neonates delivered by vaginal birth, those born by cesarean section did not demonstrate characteristic decreases in the FcGBP concentrations over the course of the third trimester. It remains to be elucidated whether these specific differences in the meconium and amniotic fluid FcGBP concentrations may offer evidence for impaired maturation of intestinal mucosal immunity in neonates born by cesarean section.

The present study was performed using specimens of amniotic fluid and meconium to assess the relationship between FcGBP and the fetal intestine. Both are biological materials that remain in direct contact with the developing fetal intestine. While changes in the composition of amniotic fluid are dynamic and representative of short intervals, accumulation of substances in meconium may reflect their presence for longer periods during gestation. Meconium is the first stool passed by the newborn. It is formed in the fetal intestine and consists of components of the amniotic fluid and the cells and secretions of the liver, pancreas, and gastro-intestinal tract [15,27].

A potential limitation of the study is the lack of the meconium and amniotic fluid IgG measurements performed in tandem with the FcGBP determinations. Further studies should also address the relationship between FcGBP levels and FcRn in the meconium and amniotic fluid. The presented studies are, according to the literature data, performed for the first time and are the basis for their continuation. Subsequent studies should take into account clinical variables such as maternal infections, condition on c-section delivery, antibiotic use, duration of labor, or prolonged rupture of membranes.

In summary, the presence of FcGBP in the meconium and amniotic fluid demonstrated for the first time in this study can start a new research direction to investigate the involvement of FcGBP in the transport of IgG and antigen-bound IgG across the fetal intestinal epithelium. Consistent decreases in the meconium FcGBP concentrations in the course of the third trimester suggest a gradually increasing interaction of fetal luminal antigens with the specific FcRn transport across the intestinal wall. Higher concentrations of FcGBP in the meconium and amniotic fluid from neonates delivered by cesarean section may be another argument confirming the hypothesis of association between disorders of the maturation of the fetal intestinal immune system and autoimmune disease in the postnatal period and later life.

## 4. Materials and Methods

### 4.1. Neonates

The study was performed on 120 neonates (58 females, 62 males) born by cesarean section (*n* = 86, 72%) and by vaginal birth (*n* = 34, 28%) in the Department of Obstetrics, Women’s Diseases, and Gynecologic Oncology at the National Medical Institute of the Ministry of the Interior and Administration (formerly Central Clinical Hospital of the Ministry of the Interior and Administration) in Warsaw. Meconium and amniotic fluid samples were collected and used in the study in compliance with the ethical standards of the Bioethical Committee at the Central Clinical Hospital of the Ministry of the Interior and Administration (Decision No. 71/2011). All deliveries were singleton and with clear amniotic fluid, and the newborns did not pass meconium inside the uterus or during delivery, but after birth. The age (mean ± SD: 39.2 ± 4.1 years) distribution of mothers was normal (Shapiro–Wilk test). 75% of the study participants were between 35 and 45 years old. The mothers were informed about the purpose and methodology of the study and signed consent.

Neonates’ characteristics:-Birth weight (mean± SD; range) [g]: 3481 ± 485; (1940–4960)-Birth length (mean± SD; range) [cm]: 55.1 ± 2.4; (47–60)-Apgar score: 10/10/10/10 (*n* = 97), 9/10/10/10 (*n* = 3), 8/9/10/10 (*n* = 3), 9/9/10/10 (*n* = 5), 9/9/9/10 (*n* = 2), 8/9/9/10 (*n* = 1), 8/8/9/10 (*n* = 1), 9/9/9/9 (*n* = 3), 9/8/9/9 (*n* = 2), 8/8/8/9 (*n* = 1), 7/8/8/9 (*n* = 1), 7/7/8/9 (*n* = 1).

### 4.2. Materials

#### 4.2.1. Meconium

First-pass meconium samples were collected. The empty tubes were weighed prior to adding meconium and reweighed after filling. Using a plastic spatula, the meconium was transferred from the nappy into a 50 mL graduated plastic tube and frozen at −20 °C for up to 7 days and next stored frozen at −80 °C until the assays were performed. The date, time, and weight of each meconium collection were recorded.

The assays were performed using the supernatant of meconium homogenate. To prepare the homogenate, PBS (phosphate-buffered saline pH 7.4) was added to a tube with meconium: one part by weight of meconium per four parts by weight of PBS. The tube was placed on a hematology mixer for 15 min and then shaken in a horizontal position using an LP300Hk shaker for 1 h. The homogenate was transferred to Eppendorf tubes and stored at 80 °C. Prior to FcGBP measurements, the homogenates were thawed for 24 h in a refrigerator.

#### 4.2.2. Amniotic Fluid

Amniotic fluid was collected vaginally with a sterile syringe after rupture of the amniotic membranes before the baby was born. Samples of amniotic fluid that were mixed with blood were excluded. The samples of amniotic fluid collected from the mothers corresponded to the samples of first-pass meconium collected from the neonates. Approximately 2–3 mL of amniotic fluid was transferred to a plastic tube and stored frozen at −80 °C until the assays were performed. Prior to FcGBP measurements, the samples were thawed and next centrifuged for 15 min at 1000 rpm.

### 4.3. Methods

FcGBP concentrations in the samples of meconium homogenates and amniotic fluid were measured using the test Human Fc Fragment of IgG Binding Protein, FcgBP ELISA Kit, Shanghai SunRedBio (8F, Block C, No. 3 Building, Zijin Plaza, No. 701, Gudun Road, Hangzhou, Zhejiang 310030, China, “https://sunredbio.guidechem.com/ (accessed on 15 May 2023)”.

The measurements were performed in duplicate in compliance with the kit manufacturer’s instructions and with the principles of Good Laboratory Practice (GLP). The mean value was calculated and reported as a final concentration in ng/mL (amniotic fluid) and in ng/g (meconium).

### 4.4. Statistical Analysis

Statistical analysis was performed using Statistica Version 13 (StatSoft Inc., TIBCO Software Inc., Palo Alto, CA, USA). The normality of distribution for the meconium and amniotic fluid FcGBP concentrations was assessed using the Shapiro–Wilk test. The non-parametric Spearman test was used to determine the correlation coefficients (r). To compare and assess the differences in the meconium and amniotic fluid FcGBP concentrations between neonates delivered by cesarean section and by vaginal birth and between neonatal sexes, the Mann–Whitney U test with the continuity correction was performed. The meconium and amniotic fluid FcGBP measurements are reported as the mean ± SD, median, range, and coefficient of variation (CV). The statistical significance level was assumed as *p* < 0.05.

## 5. Summary and Conclusions

The presence and characteristic properties of FcGBP in the meconium and amniotic fluid suggest its potential uses in medical practice as a parameter to assess the immune maturity of the fetal intestine.

Variability and a high dispersion of the meconium FcGBP concentrations in neonates born at gestational ages of 36–41 weeks may confirm the effects of individual mechanisms controlling the FcGBP function in the fetal intestine.

Amniotic fluid and meconium are biological materials obtained in the perinatal period which may be used to assess both short-term and long-term changes in the intrauterine environment.

## Figures and Tables

**Figure 1 ijms-26-07579-f001:**
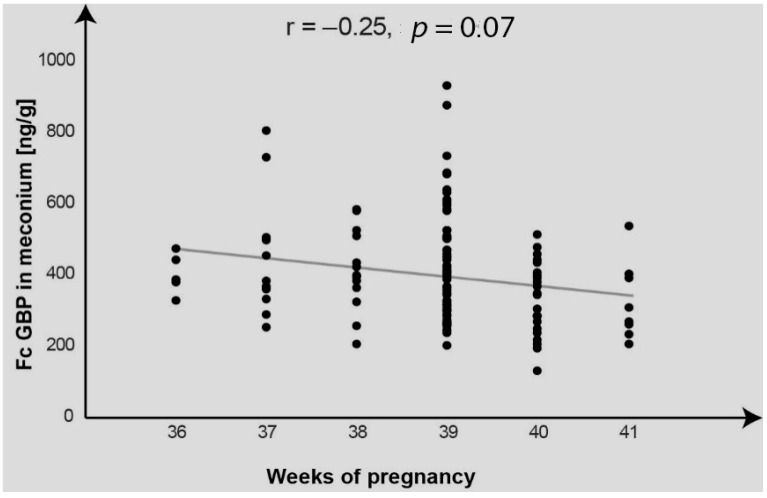
Variations in the meconium FcGBP concentrations in neonates born at 36 to 41 weeks gestation.

**Figure 2 ijms-26-07579-f002:**
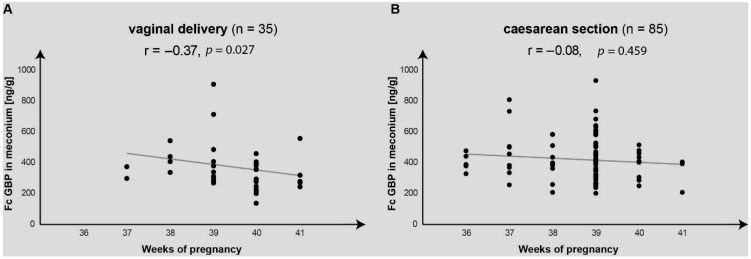
Variations in the meconium FcGBP concentrations over 36 to 41 weeks gestation. (**A**) Neonates delivered by vaginal birth. (**B**) Neonates delivered by cesarean section.

**Table 1 ijms-26-07579-t001:** FcGBP concentrations in corresponding samples of meconium and amniotic fluid.

Biological Material	FcGBP Concentration		
	Mean ± SD	Median	Range
Meconium [ng/g]	405.78 ± 145.27	388.94	134.96–933.68
Amniotic fluid [ng/mL]	135.70 ± 35.83	138.24	39.36–193.67

**Table 2 ijms-26-07579-t002:** Comparison the meconium and amniotic fluid FcGBP concentrations between newborns delivered by vaginal birth and by cesarean section.

concentration FcGBP Mean ± SD; median (range)	Delivery method	
	Vaginal birth (*n* = 35)	Cesarean section (*n* = 85)
Meconium [ng/g]	352.73 ± 142.16; 327.66 (134.96–880.13)	427.63 ± 141.62; 410.62 (202.84–933.68)
Amniotic fluid [ng/mL]	121.56 ± 38.09; 126.70 (39.36–182.51)	141.54 ± 33.42; 148.57 (61.99–196.43)

**Table 3 ijms-26-07579-t003:** The correlation coefficients between the FcGBP concentrations in the meconium and the amniotic fluid and the birth weight.

Parameter	Correlation Coefficient
Birth weight vs. meconium FcGBP	r = 0.01, *p* = 0.914
Birth weight vs. amniotic fluid FcGBP	r = 0.02, *p* = 0.871

## Data Availability

All data generated or analyzed during this study are included in this published article.

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
