# Peer review of "Meconium and Amniotic Fluid IgG Fc Binding Protein (FcGBP) Concentrations in Neonates Delivered by Cesarean Section and by Vaginal Birth in the Third Trimester of Pregnancy"

_ijms, 2025, doi:10.3390/ijms26157579_

Round 1
Reviewer 1 Report
Comments and Suggestions for Authors
Reviewer Comments
I have carefully read the manuscript entitled Meconium and amniotic fluid IgG Fc binding protein (FcGBP) concentrations in neonates delivered by cesarean section and by vaginal birth
in the third trimester of pregnancy. Overall, the study is interesting, clearly written, and addresses a relevant topic in neonatal and immunological research.
- The Introduction is well-structured, with a logical flow and a clearly defined objective.
- The Materials and Methods section is concise yet provides sufficient detail for understanding the study design and procedures.
- The Results are presented appropriately and reflect the data accurately.
- Both the Discussion and Conclusion are coherent and aligned with the reported findings.
However, I would like to pose the following questions and considerations to the authors for clarification and potential enhancement of the manuscript:
-
Did the authors take into account clinical variables such as maternal infections, antibiotic use, duration of labor, or prolonged rupture of membranes? These factors could potentially influence FcGBP levels or mucosal development.
-
What was the age distribution of the mothers? Were they all primiparous, or was there variation in parity? This information could be relevant in interpreting immunological conditions during pregnancy.
-
Amniotic fluid protein concentrations can vary with fluid volume and composition, particularly in late gestation. Was any correction made for amniotic fluid dilution, or was its concentration assessed?
-
Were any assays conducted to confirm the absence of fecal or urinary contamination in the amniotic fluid samples? Contamination could affect FcGBP concentration measurements.
-
Why was the gestational age range of 36 to 41 weeks selected? Were preterm cases (<36 weeks) deliberately excluded? If so, what was the rationale behind this decision?
-
Given that certain immune proteins can exhibit sex-specific expression, why were differences based on neonatal sex not analyzed? This could provide further insights into FcGBP function and regulation.
-
Do the authors plan to extend their research to include neonates with intestinal pathologies such as necrotizing enterocolitis? This could significantly strengthen the clinical relevance of FcGBP as a potential biomarker or functional mediator.
I believe addressing these points would further enhance the clarity, depth, and scientific robustness of the manuscript.
Author Response
Comments and Suggestions for Authors
Reviewer Comments
I have carefully read the manuscript entitled Meconium and amniotic fluid IgG Fc binding protein (FcGBP) concentrations in neonates delivered by cesarean section and by vaginal birth
in the third trimester of pregnancy. Overall, the study is interesting, clearly written, and addresses a relevant topic in neonatal and immunological research.
- The Introduction is well-structured, with a logical flow and a clearly defined objective.
- The Materials and Methods section is concise yet provides sufficient detail for understanding the study design and procedures.
- The Results are presented appropriately and reflect the data accurately.
- Both the Discussion and Conclusion are coherent and aligned with the reported findings.
However, I would like to pose the following questions and considerations to the authors for clarification and potential enhancement of the manuscript:
- Did the authors take into account clinical variables such as maternal infections, antibiotic use, duration of labor, or prolonged rupture of membranes? These factors could potentially influence FcGBP levels or mucosal development.
Answer: Thank you for your valuable advice. The presented studies are, according to the literature data, performed for the first time and are the basis for their continuation. Subsequent studies should take into account clinical variables such as maternal infections, antibiotic use, duration of labor, or prolonged rupture of membranes. (sentence placed in the Limitation of the study section)
- What was the age distribution of the mothers?Were they all primiparous, or was there variation in parity? This information could be relevant in interpreting immunological conditions during pregnancy.
Answer: The age (mean ± SD : 39.2±4.1 years) distribution of mothers was normal. 75% of the study participants were between 35 and 45 years old. (sentence placed in the Materials and Methods section).
- Amniotic fluid protein concentrations can vary with fluid volume and composition, particularly in late gestation. Was any correction made for amniotic fluid dilution, or was its concentration assessed?
Answer: Such determinations were not performed and, according to the valuable suggestion of the Reviewer, they require inclusion in further studies.
- Were any assays conducted to confirm the absence of fecal or urinary contamination in the amniotic fluid samples? Contamination could affect FcGBP concentration measurements.
Answer: Such determinations were not made. This is a very important practical note for further research.
- Why was the gestational age range of 36 to 41 weeks selected?Were preterm cases (<36 weeks) deliberately excluded? If so, what was the rationale behind this decision?
Answer: The decision to select the study group of infants aged 36-41 weeks was dictated by the need to create a group of newborns as homogeneous as possible, excluding prematurity. The obtained results encourage further study of a separate group of newborns aged <36 weeks and comparing them with term newborns.
- Given that certain immune proteins can exhibit sex-specific expression, why were differences based on neonatal sex not analyzed?This could provide further insights into FcGBP function and regulation.
Answer: There were no significant differences in FcGBP concentrations in meconium and amniotic fluid between boys (n=62) and girls (n=58), p>0.05. ( sentence placed in the results section).
- Do the authors plan to extend their research to include neonates with intestinal pathologies such as necrotizing enterocolitis?This could significantly strengthen the clinical relevance of FcGBP as a potential biomarker or functional mediator.
Answer: Yes, such studies are planned.
I believe addressing these points would further enhance the clarity, depth, and scientific robustness of the manuscript.
Submission Date
07 June 2025
Date of this review
08 Jun 2025 15:05:45
Dół formularza
© 1996-2025 MDPI (Basel, Switzerland) unless otherwise stated
Disclaimer Terms and Conditions Privacy Policy
All MDPI websites use third-party website tracking technologies to provide and continually improve our services. I agree and may revoke or change my consent at any time with effect for the future

Reviewer 2 Report
Comments and Suggestions for Authors
Thank you for the opportunity to review the article " Meconium and amniotic fluid IgG Fc binding protein concentrations in neonates delivered by cesarean section and by vaginal birth in the third trimester of pregnancy".
- Could the authors provide the mean, median, and SD of FcGBP for the same groups in Figure 1? There seems to be imbalance in the number of data for each group. Could the correlation be skewed by this data imbalance?
- Are all birth by cesarian section elective or has medical need? Please clarify in the paper.
- Why did the authors did not measure the IgG concentrations in amniotic fluidc and meconium?
- Is the assay used for FcGBP also applicable for meconium matrix? Meconium is not a homogenous matrix, therefore the sampling of meconium could affect the quantitative results. The method seems to suggest it only one sampling was done, could the authors clarify this?
Author Response
It has been revised as per the requirements.
Round 2
Reviewer 2 Report
Comments and Suggestions for Authors
Thank you for the edits. Could the authors please provide information the condition on c-section delivery? Are they all elective or due to medical necessity?
Author Response
Response to Reviewers
The presented study does not differentiate between conditions for c-section delivery. Thank you and I agree with the reviewer's comment. Differentiation between clinical causes of c-section delivery is planned in future studies. I have included information about the lack of differentiation between c-section delivery conditions in the Discussion-Limitation section of the study (changes marked in blue).
